# HSP47: A Therapeutic Target in Pulmonary Fibrosis

**DOI:** 10.3390/biomedicines11092387

**Published:** 2023-08-25

**Authors:** Noriho Sakamoto, Daisuke Okuno, Takatomo Tokito, Hirokazu Yura, Takashi Kido, Hiroshi Ishimoto, Yoshimasa Tanaka, Hiroshi Mukae

**Affiliations:** 1Department of Respiratory Medicine, Nagasaki University Graduate School of Biomedical Sciences, Nagasaki 852-8501, Japan; 2Center for Medical Innovation, Nagasaki University, Nagasaki 852-8588, Japan

**Keywords:** heat shock protein 47, pulmonary fibrosis, anti-fibrotic agent, collagen, chaperon, *SERPINH1*, fibroblast, alveolar epithelial cell

## Abstract

Idiopathic pulmonary fibrosis (IPF) is a chronic lung disease characterized by a progressive decline in lung function and poor prognosis. The deposition of the extracellular matrix (ECM) by myofibroblasts contributes to the stiffening of lung tissue and impaired oxygen exchange in IPF. Type I collagen is the major ECM component and predominant collagen protein deposited in chronic fibrosis, suggesting that type I collagen could be a target of drugs for fibrosis treatment. Heat shock protein 47 (HSP47), encoded by the serpin peptidase inhibitor clade H, member 1 gene, is a stress-inducible collagen-binding protein. It is an endoplasmic reticulum-resident molecular chaperone essential for the correct folding of procollagen. HSP47 expression is increased in cellular and animal models of pulmonary fibrosis and correlates with pathological manifestations in human interstitial lung diseases. Various factors affect HSP47 expression directly or indirectly in pulmonary fibrosis models. Overall, understanding the relationship between HSP47 expression and pulmonary fibrosis may contribute to the development of novel therapeutic strategies.

## 1. Introduction

Idiopathic pulmonary fibrosis (IPF) is a chronic fibrosing interstitial pneumonia of unknown etiology accompanied by the radiological and histological features of the usual interstitial pneumonia (UIP). It occurs primarily in older adults, presents with progressively worsening dyspnea and lung function, and has a poor prognosis [1]. The pathogenesis of fibrosis is characterized by the recurrent epithelial cell injury, senescent alveolar epithelial cells, and release of pro-fibrotic mediators, leading to the extracellular matrix (ECM) deposition by myofibroblasts [2]. These depositions result in lung tissue stiffening, which hinders lung expansion and thereby impairs oxygen exchange. Two anti-fibrotic drugs, nintedanib and pirfenidone, have been shown to be safe and effective treatments of IPF [3,4,5]. Although these drugs have distinct mechanisms of action, they both reduce the accumulation of ECM components, including collagens [2]. However, a subset of patients with interstitial lung disease (ILD) develop a progressive fibrosing phenotype similar to that of patients with IPF. This condition was termed progressive fibrosing interstitial lung disease (PF-ILD) [6] or, more recently, progressive pulmonary fibrosis (PPF) [1]. Criteria for PF-ILD and PPF include worsening respiratory symptoms, physiological evidence of disease progression, and radiological evidence of disease progression. Both of them share key processes that underpin IPF pathogenesis. The UIP pattern can be observed in PPF and PF-ILD, and it is not limited to IPF alone; the presence of the UIP pattern is associated with poor prognosis. Repeated lung injuries owing to environmental, genetic, epigenetic, and/or microbiological causes set off a cascade of events leading to abnormal wound healing. This phenomenon then evolves into a pathogenic fibrotic response in which fibroblasts differentiate into a contractile myofibroblast phenotype that accumulates in the ECM, resulting in progressive fibrosis [7]. Therefore, it has been argued that anti-fibrotic drugs may also have favorable effects on PF-ILD, in addition to their action on IPF. Indeed, the annual rate of decline in the forced vital capacity was reduced by nintedanib in patients with systemic sclerosis-associated ILD and PF-ILD [8,9]. However, anti-fibrotic drugs have major limitations, such as insufficient curative effects and poor pharmacokinetic properties. Therefore, the search for novel therapeutic targets is broadening, and the number of new drugs entering clinical trials is increasing [10].

Heat shock proteins (HSPs) are key regulators of cell homeostasis that act as chaperones for multiple proteins, including collagens. HSPs are highly conserved across species and subdivided into families based on their molecular weights [11,12]. Among them, HSP47 has been reported to be implicated in fibrotic disorders such as scleroderma, renal interstitial fibrosis, peritoneal fibrosis, cardiac fibrosis, keloid fibrosis, and pulmonary fibrosis [13]. Therefore, HSP47 has attracted attention as a target of anti-fibrotic drugs for treatment of various organs, including the lungs [14]. Here, we review the fibrotic roles of HSP47 and its potential as a treatment target in pulmonary fibrosis.

## 2. HSP47 as a Collagen-Specific Molecular Chaperone

Type I collagen is the major component of the ECM and predominant collagen deposited in chronic fibrosis, suggesting that it may be targeted for the treatment of fibrosis [15]. Type I collagen is composed of two α1 chains and one α2 chain, which assemble into triple helix superstructures within the endoplasmic reticulum (ER) and are then transported to the Golgi apparatus [16,17]. Type I collagen is thermally unstable even at body temperature, and the correct folding of procollagen requires the presence of chaperone-like cofactors [18]. HSP47 is a stress-inducible collagen-binding protein encoded by serpin peptidase inhibitor clade H, member 1 (*SERPINH1*). It is an ER-resident molecular chaperone essential for the correct folding of procollagen in vertebrate cells [19]. HSP47 binds to the Gly-Xaa-Arg motif of the triple-helical procollagen in the ER via hydrophobic and hydrophilic interactions, and prevents its local unfolding and aggregate formation, effectively chaperoning the formation of the triple helix [19]. HSP47 transiently associates with triple-helical procollagens in the ER and dissociates at the cis-Golgi, returning to the ER via its ER retention signal [20]. In vitro, purified recombinant HSP47 directly binds the triple-helical collagen with high association and dissociation rates [21]. Although many enzymes responsible for post-translational modification of procollagen bind its monomer form, HSP47 barely binds non-triple-helical procollagen. In the absence of HSP47, the rigid triple helix structure is not formed, and *Serpinh1* knockout mice do not survive beyond 11.5 days post coitus [22], indicating that HSP47 is indispensable for collagen synthesis and secretion. HSP47 shares its binding site on type I collagen with integrin α2β1 [23] and collagenase MMP [24]. Therefore, HSP47 affects integrin-mediated signaling, triple helix stability, and the turnover of collagen by collagenase. A complex network involving transforming growth factor β (TGF-β) and interleukin 1β (IL-1β), two principal fibrinogenic cytokines, induces the trimerization of heat shock transcription factor 1, allowing it to bind to the heat shock element and ultimately increase HSP47 expression [25].

## 3. Relationship between HSP47 and Pulmonary Fibrosis

### 3.1. Cellular Models

Several factors affect HSP47 expression in pulmonary fibrosis models in vitro.

Treatment with TGF-β1, an extracellular factor regulating fibrosis, stimulated both the mRNA and the protein expression levels of HSP47 and collagen type 1 in normal human fibroblasts (NHLFs) [26] and the human alveolar epithelial cell line A549 [27].

IL-1β has a pro-fibrotic role in pulmonary fibrosis [28], and it has been reported to induce HSP47 protein synthesis in human embryonic fibroblasts [25].

An imbalance between different nitric oxide synthases is thought to impact the development of lung fibrosis [29]. Exogenous NO induced the expression of HSP47, collagen type 1, TGF-β1, and phosphorylated SMAD-2 in human fetal lung fibroblast MRC-5 cells, suggesting that NO promotes the progression of pulmonary fibrosis via the TGF-β1/SMAD signaling cascade and increases HSP47 expression in pulmonary fibroblasts [30,31].

Human neutrophil peptides (HNPs), known as α-defensins, are the most abundant neutrophilic proteins involved in innate and acquired immunity [32]. HNPs are present in the alveolar septa, especially in the dense fibrotic area of IPF, and they have a fibrotic effect on lung fibroblasts [33,34]. HNPs increased both the mRNA and the protein expression levels of HSP47 and collagen type 1 in NHLFs [35]. 

MicroRNAs (miRNAs) are small non-coding RNAs that play pivotal roles in the regulation of the expression of protein-coding and non-coding RNAs. *miR-29a* is downregulated in both fibrotic and cancerous tissues obtained from patients with lung cancer and IPF. Gene expression data and in silico analyses showed that *SERPINH1* is a direct target of *miR-29a*. Restoration of *miR-29a* decreased HSP47 expression, whereas downregulation of *miR-29a* caused the overexpression of HSP47 in the human cancer cell lines EBC-1 and A549, and in lung fibroblast MRC-5 cells [36]. These results suggest that *miR-29a* influences the progression of pulmonary fibrosis by increasing HSP47 expression in pulmonary fibroblasts.

### 3.2. Animal Models

A close relationship between HSP47 and pulmonary fibrosis was first reported in a rat bleomycin-induced pulmonary fibrosis model approximately 25 years ago [37]. Strong immunostaining for HSP47 is observed in the bleomycin-treated fibrotic lungs, particularly in the abundant, phenotypically altered myofibroblasts and fibroblasts. In addition, parallel increases in collagen expression levels were observed in bleomycin-treated lungs and colocalization of HSP47 and collagens was detected by double immunostaining [37]. Ishii et al. also reported the expression of HSP47 protein and *Serpinh1* mRNA in murine bleomycin-induced pulmonary fibrosis by using immunohistochemical analysis and semi-quantitative RT-PCR [38]. As in the previous study, immunohistochemical analysis showed higher HSP47 expression levels in animals with bleomycin-induced pulmonary fibrosis than in controls. HSP47 was localized predominantly in α-smooth muscle actin (αSMA)^+^ myofibroblasts, surfactant protein (SP)-A^+^ type II alveolar epithelial cells, and F4/80^+^ macrophages. RT-PCR also demonstrated an increase in *Serpinh1* mRNA expression, and the relative amount of *Serpinh1* mRNA correlated significantly with lung hydroxyproline content, an indicator of pulmonary fibrosis in bleomycin-treated lungs. In situ hybridization also showed that the signal for *Serpinh1* mRNA was markedly increased in bleomycin-treated lungs compared with that in controls. *Serpinh1* mRNA was localized in αSMA^+^ myofibroblasts, SP-A^+^ type II alveolar epithelial cells, and F4/80^+^ macrophages in active fibrotic areas [39]. These reports also indicate that HSP47 is expressed in myofibroblasts, type II alveolar epithelial cells, and macrophages, i.e., in types of cells associated with lung fibrosis in animal models.

Aging is a significant risk factor for fibrosis, and IPF is characteristically associated with advanced age. Collagen deposition and the number of HSP47^+^ cells were increased in old bleomycin-treated mice compared to those in young bleomycin-treated mice [40]. 

The unfolded protein response (UPR) is a direct consequence of cellular ER stress and a key disease-driving mechanism in IPF. The resolution of the UPR is directed by PPP1R15A, the expression of which is markedly reduced in lung tissues affected by IPF. Sephin1, a small-molecule inhibitor of PPP1R15A, increased collagen 1a and *Serpinh1* mRNA expression levels in mice with bleomycin-induced pulmonary fibrosis compared to the corresponding levels in control mice [41]. These results indicate that UPR in general or the UPR regulator PPP1R15A in particular may be targets of novel therapeutics for IPF that would act through HSP47.

In contrast, HSP47 protein levels have been reported to be decreased in isolates from the lungs of rats with septic and nephrogenic acute respiratory distress syndrome, characterized histopathologically by a diffuse alveolar damage (DAD) pattern [42]. DAD is characterized by the excessive production of pro-inflammatory cytokines and chemokines, massive infiltration of neutrophils to the lungs, endothelial dysfunction, microthromboses, interstitial and alveolar edema, death of alveolar epithelial cells, and activation of macrophages, leading to severe respiratory failure due to diffuse damage to the alveolar capillary membranes [43].

### 3.3. Human Studies

#### 3.3.1. Idiopathic Pulmonary Fibrosis

The various human studies on HSP47 expression are shown in Table 1. HSP47 expression in various pulmonary fibrotic diseases was first reported by Razzaque et al. [44]. Lung sections from autopsies of 17 patients with various pulmonary fibrotic diseases, including IPF, were stained with monoclonal antibodies for HSP47. Compared with the control lung sections, markedly increased immunostaining for HSP47 was noted in the fibrotic lung sections, in association with increased accumulation of type III collagen in fibrotic masses. Double immunostaining revealed the colocalization of collagens and HSP47 in the regions of pulmonary fibrosis, and HSP47-expressing cells were found to be mainly αSMA^+^ myofibroblasts. In addition to the data from autopsy sections, Iwashita et al. reported a higher expression of HSP47 and type I procollagen in patients with IPF by using immunohistochemistry on sequential sections compared to expression in cryptogenic organizing pneumonia (COP), which is considered a weak type of fibrosis compared with IPF among idiopathic interstitial pneumonias (IIPs). HSP47 and type I procollagen are localized predominantly in αSMA^+^ myofibroblasts and SP-A^+^ type II alveolar epithelial cells in active fibrotic areas of UIP but are not or are weakly expressed in these cells in COP sections and normal lung tissue samples obtained from excised lung cancer tissues [45].

#### 3.3.2. Non-Idiopathic Pulmonary Fibrosis

The HSP47 expression level was examined in interstitial pneumonias of various etiologies characterized by distinct pathological patterns, such as idiopathic UIP (I-UIP), collagen vascular disease-associated UIP (CVD-UIP), and idiopathic nonspecific interstitial pneumonia (I-NSIP) [46]. Histologically, I-NSIP shows alveolar and interstitial mononuclear cell inflammation and fibrosis in a temporally uniform pattern, with preserved underlying alveolar architecture, and prognosis is favorable compared with IPF [47]. Patients with I-UIP, CVD-UIP, and I-NSIP had significantly higher HSP47 expression in fibroblasts than in control lung tissues. Further, patients with I-UIP and I-NSIP showed a significantly higher HSP47 expression in fibroblasts than those with CVD-UIP. Patients with I-UIP and I-NSIP had a significantly higher expression of HSP47 in type II pneumocytes than controls, whereas there was no significant difference in HSP47 expression between the CVD-UIP and control groups. HSP47 expression in type II pneumocytes of patients with I-UIP was significantly higher than that in patients with CVD-UIP and I-NSIP. The HSP47 and type I procollagen expression levels were similar in type II pneumocytes [46].

HSP47 expression in lung fibroblasts was reported to correlate with the prognosis in fibrotic NSIP [48]. There was no significant difference in HSP47 expression between idiopathic and CVD-associated NSIP. However, patients with idiopathic fibrotic NSIP that had higher HSP47 expression levels had a poorer prognosis than patients with lower HSP47 expression levels. 

Serum HSP47 levels were also evaluated in several fibrotic lung diseases. In patients with acute interstitial pneumonia characterized by the DAD pattern [49], serum HSP47 levels were significantly higher than those in patients with other IIPs, such as COP, NSIP, or IPF, and healthy subjects. In contrast, serum HSP47 levels in patients with COP, NSIP, and IPF did not differ significantly and were similar to those in healthy subjects [50]. In addition, serum HSP47 levels in patients with acute exacerbation of IPF, showing the DAD pattern superimposed on the UIP pattern [51], were significantly higher than those in patients with stable IPF. Immunohistochemical analysis revealed that pulmonary HSP47 expression was higher in DAD tissues of acute exacerbation of IPF than in UIP tissues of stable IPF [52]. Serum HSP47 levels were also examined in individuals with drug-induced lung diseases. Patients showing the DAD pattern in chest high-resolution computed tomography had worse outcomes and higher serum HSP47 levels than patients with other radiological patterns. Furthermore, receiver operating characteristic curves revealed that HSP47 was superior to other ILD markers, such as KL-6, SP-A, and SP-D, in discriminating between the groups with and without DAD [53]. DAD is characterized by severe inflammation, tissue destruction, and alveolar epithelial and endothelial injury with increased vascular permeability [54]. The results of the abovementioned studies indicate that high HSP47 expression and increased permeability because of tissue destruction caused by inflammation are the reasons for HSP47 antigen detection in the serum. In addition, HSP47 levels in the plasma and bronchoalveolar lavage fluid were significantly increased in patients with acute respiratory distress syndrome that presented with a histopathological DAD pattern, compared to the corresponding levels in patients without this syndrome [55].

In addition to the HSP47 antigen, serum anti-HSP47 autoantibody level has also been reported as a marker of fibrotic lung diseases. HSP47 antigen and anti-HSP47 autoanti-body levels detected by the enzyme-linked immunosorbent assay are significantly elevat-ed in the sera of patients with rheumatic autoimmune disease, but not in the sera of pa-tients with IPF. However, particularly high levels of the HSP47 antigen and autoantibod-ies against HSP47 were noted in the sera of patients with rheumatic autoimmune disease and mixed connective tissue disorder. In such patients, the simultaneous occurrence of systemic inflammation and upregulation of HSP47 expression likely causes leakage of HSP47 from fibrotic lesions into the peripheral blood, and the leaked antigen induces a high titer of autoantibodies to HSP47 [56]. Besides their occurrence in patients with IPF, autoantibodies against HSP47 were also detected in patients with IIPs, including I-NSIP and COP. Serum levels of these autoantibodies in patients with I-NSIP were significantly higher than in patients with IPF or COP and healthy subjects [57]. In this case, similar to the results of the previous study [56], tissue destruction caused by inflammation may ex-plain the appearance of HSP47 antigens in the serum and subsequent production of an-ti-HSP47 autoantibodies.

**Table 1 biomedicines-11-02387-t001:** Studies of HSP47 expression in human tissues.

Pathology	Sample	Method	Description	Refs.
IPF	Lung tissue	IHC	Patients with fibrotic I-NSIP with higher HSP47 expression levels had a poorer prognosis than patients with lower HSP47 expression levels	[48]
AE-IPF	Serum	ELISA	Serum HSP47 levels were significantly higher than stable IPF	[52]
AE-IPF	Lung tissue	ELISA	HSP47 expression was higher in DAD tissues of acute exacerbation of IPF than in UIP tissues of stable IPF	[52]
AIP	Serum	ELISA	Serum HSP47 levels in patients with AIP were significantly higher than those in patients with other IIPs and healthy subjects	[49]
AIP	Tissue	IHC	HSP47 expression was higher in AIP than in IPF	[49]
DILD	Serum	ELISA	Patients showing DAD pattern in chest HRCT had higher serum HSP47 level	[53]
ARDS	Plasma/BALF	Bradford protein assay	HSP47 levels in the plasma and BALF were significantly increased in patients with ARDS that presented with a histopathological DAD pattern, compared to the corresponding levels in patients without this syndrome	[55]
I-NSIP	Serum	ELISA	Serum levels of anti-HSP47 autoantibodies in patients with I-NSIP were significantly higher than in patients with IPF or COP and healthy subjects	[57]

AIP, acute interstitial pneumonia; ARDS, acute respiratory distress syndrome; BALF, bronchoalveolar lavage fluid; CT, computed tomography; CVD-UIP, collagen vascular disease-associated usual interstitial pneumonia; DAD, diffuse alveolar damage; DILD, drug-induced lung disease; ELISA, enzyme-linked immunosorbent assay; IHC, immunohistochemistry; I-NSIP, idiopathic nonspecific interstitial pneumonia; IPF, idiopathic pulmonary fibrosis; I-UIP, idiopathic usual interstitial pneumonia; αSMA, α-smooth muscle actin; UIP, usual interstitial pneumonia.

#### 3.3.3. Lung Cancer

IPF is an independent risk factor for lung cancer (LC), which accounts for approximately 10% of deaths in patients with IPF [58,59]. Even the earliest findings of pulmonary fibrosis, called interstitial lung abnormalities, increase the risk of cancer [60]. Multiple common genetic, molecular, and cellular processes that connect lung fibrosis with LC, such as myofibroblast/mesenchymal transition, myofibroblast activation and uncontrolled proliferation, endoplasmic reticulum stress, alterations in growth factor expression levels, oxidative stress, and large genetic and epigenetic variations, predispose patients to develop IPF and LC [61]. 

The altered expression levels of HSP47 have been correlated with several types of cancer, such as cervical, breast, pancreatic, and gastric cancers [62]. *SERPINH1* has been reported to be highly expressed in lung adenocarcinoma tissue, and the *SERPINH1* low-expression group had a higher survival rate than the high-expression group [63]. In addition, HSP47 expression in squamous cell carcinoma cells was shown to be higher than that in normal human bronchial epithelium cells [64]. 

The aberrant expression of the *SERPINH1* gene has been shown to be closely linked to tumor growth, invasion, and metastasis in cervical squamous cell carcinoma and gastric cancer [65,66]. In non-small cell lung cancer cell lines, HSP47 was highly expressed, whereas the inhibition of HSP47 repressed cell migration and invasion by diminishing the AKT signal [67]. 

Fibroblasts in solid tumors are known as carcinoma-associated fibroblasts. Based on paracrine and juxtracrine signals, these stromal cells modulate cancer progression both directly and indirectly [68]. Collagen is the major component of the tumor microenvironment and participates in cancer fibrosis. Collagen biosynthesis can be regulated by cancer cells through mutated genes, transcription factors, signaling pathways, and receptors, and collagen can influence tumor cell behavior through integrins, discoidin domain receptors, tyrosine kinase receptors, and signaling pathways [69]. Notably, we have reported that lung cancer patients with a high number of HSP47^+^ fibroblasts in the cancer stroma had shorter disease-free survival than those with a low number of such fibroblasts, whereas HSP47 expression in the cancer cells did not correlate with survival [70]. HSP47^+^ fibroblasts in cancer stroma may represent carcinoma-associated fibroblasts. Thus, HSP47 may play a pivotal role in the development of LC with pulmonary fibrosis and might be a therapeutic target of LC with pulmonary fibrosis. 

## 4. Changes in HSP47 Expression Caused by Therapeutics against Pulmonary Fibrosis

### 4.1. Nintedanib

Nintedanib is an intracellular inhibitor that targets multiple tyrosine kinases, including vascular endothelial growth factor, fibroblast growth factor, and platelet-derived growth factor receptors [71]. Nintedanib is approved for the treatment of IPF and other chronic fibrosing ILDs with a progressive phenotype and systemic sclerosis-associated ILD [3,8,9]. With regard to the mechanisms of nintedanib’s action, it has been reported to inhibit growth factor-induced proliferation of lung fibroblasts in vitro [72], fibroblast motility [73], and fibroblast to myofibroblast differentiation [74]. Nintedanib also reduced the TGF-β1-induced mRNA expression and secretion of collagen type 1 in lung fibroblasts [72]. These are thought to be the mechanisms of nintedanib’s antifibrotic action. The relevance of its mechanism of action for pulmonary fibrosis is unclear; nintedanib has been reported to reduce *SERPINH1* mRNA levels but not HSP47 protein levels in TGF-β1–treated IPF fibroblasts in vitro [75].

### 4.2. Pirfenidone

Pirfenidone is a bioavailable, orally administered synthetic molecule, and is beneficial in patients with IPF as it reduces disease progression and lowers mortality [4,76]. Pirfenidone has been shown to have this effect on the other chronic fibrosing ILDs and also reduce the fibrosis of different organs: the kidney, liver, heart, and vascular remodeling [77]. The anti-fibrotic mechanism of pirfenidone is associated with the inhibition of the production or activity of TGF-β1, tumor necrosis factor-alpha, fibroblast growth factor, IL-1β, and platelet-derived growth factor PDGF. Additionally, pirfenidone possesses anti-inflammatory, anti-oxidant, and anti-apoptotic qualities [77]. Pirfenidone significantly inhibited increases in both mRNA and protein expression levels of HSP47 and collagen type I stimulated by TGF-β1 in NHLF and the human alveolar epithelial cell line A549 in vitro [26,27]. In addition, treatment with pirfenidone significantly reduced the number of HSP47^+^ type II pneumocytes and fibroblast-like cells [78]. These results indicate that pirfenidone exerts an anti-fibrotic effect by suppressing HSP47 expression in fibrotic lungs. In contrast, another study reported that pirfenidone reduced *SERPINH1* mRNA expression, but not HSP47 protein levels, in pulmonary fibroblasts [75].

### 4.3. Diallyl Sulfide

Garlic contains diallyl sulfide and related organosulfur compounds that have anti-oxidant, anti-tumorigenic, antibiotic, and detoxifying properties [79]. Immunohistochemical staining revealed that diallyl sulfide treatment decreased the upregulated expression of HSP47 and α-SMA in the lung tissue homogenates of bleomycin-treated rats [80].

### 4.4. Emodin

Emodin is a biologically active substance that is abundant in the extract of rhubarb, a plant with long, sour-tasting, red and green stems that can be cooked and eaten. Emodin has diverse biological actions, including laxative, immunosuppressive, anti-cancer, and anti-inflammatory effects [81]. It has been reported that emodin significantly relieved fibrotic changes and reduced TGF-β1 and HSP47 levels in the lungs of bleomycin-treated rats [82].

### 4.5. Aminoguanidine

Advanced glycation end products (AGEs) bind to a specific AGE receptor and activate a variety of downstream pro-inflammatory signaling cascades that converge with other factors to exacerbate fibrosis [83]. AGE levels in rat lungs, as well as lung hydroxyproline content and fibrosis score, were significantly enhanced by bleomycin stimulation, and this effect was abrogated by aminoguanidine, a crosslink inhibitor of AGE formation. Bleomycin significantly increased *Serpinh1* mRNA and HSP47 protein expression levels in rat lung tissues, whereas aminoguanidine treatment markedly decreased bleomycin-induced HSP47 expression [84].

### 4.6. Sirtuin 3

Sirtuin 3, a nicotinamide adenine dinucleotide-dependent mitochondrial deacetylase, positively modulates many cellular processes, including energy metabolism, mitochondrial biogenesis, and anti-oxidant activity [85]. Sirtuin 3-deficient mice were found to be susceptible to bleomycin-induced pulmonary fibrosis, and HSP47 expression in the mutant animals was higher than in wild-type mice [86].

### 4.7. HSPB5

Small heat shock protein 5 (HSPB5), also called αB-crystallin (αB-c), is among the 10 most common human HSPBs [87]. Intracellularly, HSPB5 serves as a molecular chaperone, being the first line of defense and preventing target protein unfolding and aggregation under cellular stress conditions [88]. The interactions between desmin and actin suggest that αB-crystallin may be important for the cytoskeletal architecture of fibroblasts and myofibroblasts during fibrogenesis [14]. HSPB5 knockout mice were protected from bleomycin-induced fibrosis. The increase in HSP47 expression induced by bleomycin in HSPB5 KO mice was much lower than in WT mice [89].

### 4.8. Corticosteroids

Asthma and chronic obstructive pulmonary disease are common chronic inflammatory diseases that also present with fibrosis, although its localization may differ from that observed in IPF. Budesonide and fluticasone are widely used inhaled steroids. It has been reported that these drugs downregulated collagen and *SERPINH1* mRNA expression in primary human lung fibroblasts in vitro [90].

### 4.9. SERPINH1 siRNA

*SERPINH1* small interfering RNA (siRNA) has attracted attention as a potential therapeutic agent, as it dose-dependently decreased collagen type I expression in TGF-β1-treated lung fibroblasts [91]. In addition, treatment with siRNA against *Serpinh1* encapsulated in vitamin A-coupled liposomes potently suppressed HSP47 expression and induced the apoptosis of myofibroblasts in the lungs of bleomycin-treated rats. Morphological pulmonary fibrosis, hydroxyproline levels, inflammatory cytokines in the lungs, and the number of inflammatory cells in the bronchoalveolar lavage fluid of bleomycin-treated rats were significantly suppressed by *Serpinh1* siRNA [92]. In another study, drug ND-L02-s0201, representing *Serpinh1* siRNA encapsulated in lipid nanoparticles, dose-dependently and statistically significantly reduced the relative lung weight, collagen deposition, and fibrosis scores in bleomycin-treated rats. Ex vivo evaluations showed that bleomycin administration increased the number of myofibroblasts, whereas treatment with ND-L02-s0201 reduced it. Comparable anti-fibrotic efficacy of ND-L02-s0201 was also observed in the silica-induced pulmonary fibrosis model [93]. A phase II study evaluating the safety, biological activity, and pharmacokinetics of ND-L02-s0201 in patients with IPF is in progress [94]. The therapeutic potential of *Serpinh1* siRNA was evaluated in fibrogenic precision-cut lung slices prepared from murine tissue. TGF-β1 promoted mRNA expression and the secretion of collagen type I as well as the transcription of other fibrogenesis-related mRNAs. After silencing *Serpinh1,* only fibronectin secretion was reduced, whereas other aspects of fibrogenesis, including the production of collagen type I, remained unaffected [95]. 

### 4.10. Other Compounds Inhibiting HSP47 Functions 

A small molecule that blocks the collagen chaperone function of HSP47 and inhibits collagen fibril formation has been described recently [96]. This HSP47 inhibitor decreased collagen type I expression in TGF-β1-treated lung fibroblasts in a dose-dependent manner. This inhibitor also decreased the viability and migration ability of TGF-β1-treated lung fibroblasts [91]. 

Using the collagen fibril formation screening assay, epigallocatechin-3-O-gallate, a typical polyphenol compound derived from tea leaves, was identified as an HSP47 inhibitor. Structurally related compounds have been synthesized and examined for their activity, revealing that the hydroxyl group at position 5 of the A-ring is important for the inhibitory activity toward HSP47 [97]. 

Ito et at. found a small-molecule compound that inhibited the interaction of HSP47 with collagen in chemical libraries using surface plasmon resonance. Compound Col003 competitively inhibited this interaction and suppressed collagen secretion by destabilizing the collagen triple helix. Structural information obtained by nuclear magnetic resonance analysis revealed the competitive binding of Col003 to the collagen binding site on HSP47 [98]. Subsequently, they synthesized Col003 via Pd(0)-catalyzed cross-coupling reactions of 5-bromosalicylaldehyde derivatives with alkyl metal species. Two potent inhibitors were discovered that inhibited the interaction between collagen and HSP47 by 85% and 81%, respectively, at a concentration of 1.9 µM [99].

## 5. Problems and Challenges with HSP47-Targeted Therapeutic Drugs

In general, small-molecule inhibitors offer advantages in terms of drug-like properties, oral administration, and bioavailability. However, researchers face challenges in designing highly specific inhibitors which may lead to potential side effects. In particular, when considering inhibitors that target HSP47, HSP47 is anticipated to cause side effects due to its inhibition of collagen production. Collagen is a ubiquitous molecule in the body and fulfills essential functions as a macromolecular scaffold, growth factor reservoir, and receptor binding site in virtually every tissue [100]. Therefore, inhibition of HSP47 might lead to adverse connective tissue alternations. In addition, mutations in *SERPINH1* have been reported to cause osteogenesis imperfecta [101]. This indicates that targeting HSP47 might affect normal bone development. In addition, HSP47 interacts with inositol-requiring enzyme-1alpha, a regulator of the unfolded protein response during ER stress [102]. Targeting HSP47 to treat fibrotic diseases raises concerns about potential ER stress and accumulation of misfolded proteins. ER stress-induced apoptosis of alveolar epithelial cells contributes to fibrosis development [103]. Targeting HSP47 for antifibrotic therapy might not be suitable due to the potential activation of ER stress and fibrotic processes [100].

Given the side effects of small-molecule inhibitors, a therapy that suppresses HSP47 only at sites of particularly high HSP47 expression in pulmonary fibrosis may be preferable. A new interdisciplinary area of study combines nanotechnology and medicine. An application of nanotechnology in medicine is called nanomedicine, while the pharmaceutical products containing nanotechnology are known as nanomedicines [104]. Specifically, in terms of drug delivery, nanomedicine is promising in that the use of nanomaterials can improve the delivery of poorly soluble drugs in water, targeted delivery, and long release of drugs, and in general terms, improve the overall drug performance [105]. A variety of nanostructures exist that are used for drug delivery purposes; these nanostructures can vary in shape, size, and material used. The variation among the variables can modify the behavior and functionalities of the structure within the biological environment, the drug that the structure transports; thus, the materials used can be combined to improve the structure’s properties [106]. Research in pulmonary medicine has reported on several nanostructures, such as polymeric nanoparticles, polymeric micelles, liposomes, and lipid-based nanoparticles. In clinical trials, liposomes were the principal nanostructures, which have been reported by other studies due to their biocompatibility and easy fabrication methods. Liposomes are followed by exosomes, where mesenchymal stem cells-derived exosomes are being studied to treat inflammation and promote regeneration produced in diseases such as the new COVID-19 variants or bacterial pulmonary infections [107]. 

Although no treatment for pulmonary fibrosis targeting HSP47 is yet available to the public, it is hoped that the above advantages and disadvantages will be considered in drug development.

## 6. Conclusions

HSP47 expression is increased in cellular and animal models of pulmonary fibrosis. The HSP47 expression level correlates with disease parameters in human interstitial lung diseases. Various factors directly or indirectly affect HSP47 expression in pulmonary fibrosis (Figure 1). Overall, understanding the relationship between HSP47 expression and pulmonary fibrosis may contribute to the development of novel therapeutic strategies. HSP47 may be a promising target for controlling pulmonary fibrosis because it regulates collagen synthesis and fibrogenesis. On the contrary, there are still many challenges in validating the effectiveness of a therapeutic drug targeting HSP47 for the treatment of pulmonary fibrosis. The first challenge is the development of drugs that can efficiently inhibit the overexpression of HSP47 at the specific sites of fibrotic lesions in the lungs and the issue of drug delivery. The second challenge is the potential for off-target effects resulting from the selective inhibition of HSP47. Despite these issues, which need to be addressed in future research, HSP47-targeted therapy is considered a promising approach for treating pulmonary fibrosis and is a worthwhile endeavor.

## Figures and Tables

**Figure 1 biomedicines-11-02387-f001:**
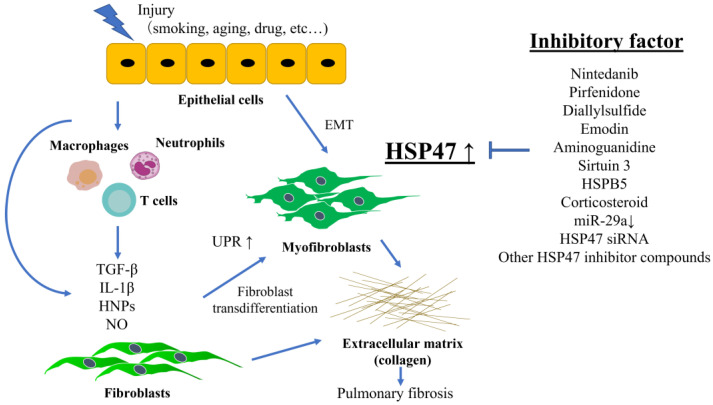
Relationship between HSP47 and pulmonary fibrosis, as well as reported factors directly or indirectly affecting HSP47 expression in pulmonary fibrosis.

## Data Availability

No new data were created or analyzed in this study. Data sharing is not applicable to this article.

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
