# Peer review of "HSP47: A Therapeutic Target in Pulmonary Fibrosis"

_biomedicines, 2023, doi:10.3390/biomedicines11092387_

Round 1

Reviewer 1 Report

I would like to thank the handling editor for offering me the opportunity to review the manuscript entitled “HSP47: Therapeutic Target in Pulmonary Fibrosis” authored by Sakamoto and colleagues, which is currently under consideration for publication in Biomedicines. I would also like to commend the authors for their scholarly work, which presents a narrative review that examines the role of heat shock protein 47 (HSP47) in pulmonary fibrosis and its potential as a therapeutic target. HSP47 is a collagen-specific molecular chaperone that is essential for collagen folding and secretion. The authors describe how HSP47 expression is upregulated in cellular and animal models of pulmonary fibrosis and correlates with disease severity in human lung fibrotic disorders, such as idiopathic pulmonary fibrosis (IPF). Various cytokines, molecules, and conditions are reported to induce HSP47 expression directly or indirectly in lung fibroblasts and epithelial cells. The review discusses studies showing increased HSP47 levels in IPF lungs and serum samples of patients, especially those with acute exacerbations. Anti-fibrotic drugs, such as nintedanib and pirfenidone, are found to reduce HSP47 expression in lung cells. Other compounds that inhibit HSP47 expression or collagen chaperone function also ameliorate experimental lung fibrosis. RNAi-based HSP47 silencing is highlighted as a promising therapeutic approach currently in clinical trials. Overall, the review makes a case for HSP47 being a key pro-fibrotic factor and a potential target for developing anti-fibrotic therapies in pulmonary fibrosis.

This review provides a timely and comprehensive overview of the current evidence linking HSP47 to pulmonary fibrosis pathogenesis. The topic is relatively novel, as no recent reviews have extensively focused on the fibrotic roles of HSP47 and its potential as an anti-fibrotic target, despite emerging data in this area. By collating findings from cellular, animal, and human studies, the authors make a convincing case for HSP47 being a key pro-fibrotic molecule that correlates closely with disease severity. The review adds value to the literature by highlighting the various cytokines, drugs, and conditions that regulate HSP47 expression in lung cells. This analysis strengthens the rationale for therapeutically targeting HSP47 in pulmonary fibrosis. The ongoing clinical development of HSP47 inhibitors is also noteworthy.

Overall, the manuscript is well-written and structured logically to synthesize the current state of the field. The topic is well-timed and aligned with increasing interest in antifibrotic therapies for progressive fibrosing lung diseases. By underscoring the preclinical and clinical evidence linking HSP47 to fibrosis pathology, this review could impact the field by stimulating further research into HSP47-based antifibrotic approaches. The authors have identified a significant knowledge gap and their work provides a framework for future studies aimed at validating HSP47 as a prognostic biomarker or drug target in pulmonary fibrosis.

While the manuscript provides valuable insights, there are several areas that could be refined to further augment the quality and impact of the work. Here are some respectful suggestions:

·         The authors could consider expanding the section on HSP47's roles in lung cancer associated with pulmonary fibrosis. This relationship is interesting but covered only briefly currently. Elaborating on the mechanisms linking HSP47 to both fibrosis and cancer progression may make this section more impactful.

·         Additional figures summarizing the key cellular sources, inducers, and inhibitors of HSP47 in pulmonary fibrosis models could further aid reader comprehension. Visual abstract-style diagrams are an effective way to communicate complex mechanisms.

·         The conclusion could be strengthened by providing more perspective on future directions, unanswered questions and challenges that need to be addressed before HSP47-targeted therapies can be clinically realized. This will help situate the current evidence and underscore gaps requiring further research.

·         Given the focus on HSP47 as a drug target, more comparison of emerging small molecule versus RNAi-based approaches may be warranted in the therapeutic sections. The pros and cons of each could be discussed.

·         The section on human studies of HSP47 in pulmonary fibrosis is extensive. The authors could consider structuring it more clearly, perhaps by separating findings in IPF vs non-IPF fibrotic lung diseases.

·         Tables summarizing the various human studies on HSP47 expression, serum levels, autoantibodies, etc. may help condense the results and highlight key takeaways for readers.

·         Discussion of limitations of targeting HSP47, such as potential side effects or challenges with delivery, could add balance and identify open questions needing study.

·         The authors could ensure vocabulary and language is accessible to a broad scientific audience by explaining specialized terms in pulmonary biology/medicine (e.g., UIP, DAD, PF-ILD) on first use.

In conclusion, I would like to reiterate my appreciation to both the editor and the authors for the opportunity to review this intriguing and informative manuscript. I trust that my suggestions will help enhance the rigor, clarity, and relevance of this important work. I look forward to seeing the revised version of the manuscript and wish the authors success in their ongoing research endeavours.

The English in the manuscript is quite good overall. The writing is clear and comprehensible. There are no major grammatical or spelling errors that impede understanding. The scientific writing style and vocabulary suits the topic and target journal. Some suggestions to further improve the English quality:

·         The sentence structure could be simplified in a few long, complex sentences to enhance readability.

·         There is occasional overuse of descriptive language and adverbs that could be made more concise.

·         There are minor typos in a few words that need fixing.

Reviewer 2 Report

In this review, the authors systematically review the fibrotic roles of HSP47 and its potential as a treatment target in pulmonary fibrosis. They first summarize the collagen-specific molecular chaperone function of HSP47 in pulmonary fibrosis, followed by a detailed description of its relationship with pulmonary fibrosis. They also briefly summarize the therapeutics targeted HSP47 against pulmonary fibrosis. Overall, this review is comprehensive and well-organized and should be published in Biomedicines after some minor revision.

1.       I strongly suggest the authors use a schematic graph to state the relationship between HSP47 and pulmonary fibrosis.

2.      Considering nintedanib and pirfenidone as the main anti-fibrotic drugs, it will be better to detailly to summarize their anti-fibrotic mechanism and progress in research and clinical applications.

3.      Nanomedicine also has promising potential for anti-fibrotic function and should be summarized in this review.

Minor editing of the English language required

Round 2

Reviewer 1 Report

I want to express my appreciation for the attention and consideration you have devoted to my suggested revisions for your manuscript. It is evident that a significant amount of effort and thought has been directed towards the refining of your work, integrating the feedback provided during the peer review process. The resulting modifications demonstrate a thorough and thoughtful approach, and significantly enhance the rigor and overall quality of your manuscript. I look forward to witnessing the impact your research will undoubtedly have on the academic community.